# Verteporfin Inhibits the Progression of Spontaneous Osteosarcoma Caused by Trp53 and Rb1 Deficiency in Ctsk-Expressing Cells via Impeding Hippo Pathway

**DOI:** 10.3390/cells11081361

**Published:** 2022-04-16

**Authors:** Yang Li, Shuting Yang, Shuying Yang

**Affiliations:** 1Department of Basic & Translational Sciences, School of Dental Medicine, University of Pennsylvania, Philadelphia, PA 19104, USA; liyang86@upenn.edu (Y.L.); yshut@upenn.edu (S.Y.); 2Center for Innovation & Precision Dentistry, School of Dental Medicine, School of Engineering and Applied Sciences, University of Pennsylvania, Philadelphia, PA 19104, USA; 3The Penn Center for Musculoskeletal Disorders, School of Medicine, University of Pennsylvania, Philadelphia, PA 19104, USA

**Keywords:** osteosarcoma, Trp53, Rb1, YAP/TAZ, verteporfin

## Abstract

Osteosarcoma is the most common primary malignancy of bone in children and adolescents. Others and our previous studies have shown that Yes-associated protein (YAP)/transcriptional coactivator with PDZ-binding motif (TAZ) as core components of the Hippo pathway are crucial regulators of osteosarcoma formation and progression. Recent studies demonstrated that verteporfin (VP) is an inhibitor of YAP/TAZ signaling in xenograft osteosarcoma. However, whether VP can inhibit primary osteosarcoma in mice is unknown. Mutations of Trp53 and Rb1 occur in approximately 50~70% of human osteosarcoma. In this study, we successfully generated the Ctsk-Cre;Trp53^f/f^/Rb1^f/f^ mice in which Trp53/Rb1 was ablated in Ctsk-expressing cells and found that Ctsk-Cre;Trp53^f/f^/Rb1^f/f^ mice spontaneously developed osteosarcoma with increased expansive osteoid lesions in the cortical bone with aging. Loss of Trp53/Rb1 in Ctsk-expressing cells significantly promoted the expression and nuclear translocation of YAP/TAZ. Micro-CT results showed that inhibition of YAP/TAZ by VP delays osteosarcoma progression and protected against bone erosion in Ctsk-Cre;Trp53^f/f^/Rb1^f/f^ mice. Importantly, the Kaplan–Meier survival curves displayed a significantly longer survival rate after VP treatment in Ctsk-Cre;Trp53^f/f^/Rb1^f/f^ mice compared to non-injected groups. In vitro studies further showed that VP inhibited the proliferation, migration and invasion in Trp53/Rb1-mutant Ctsk-expressing cells. Moreover, the results from promoter luciferase activity analysis showed that the transcriptional activity of YAP/TAZ was significantly increased in osteosarcoma tissue from Ctsk-Cre;Trp53^f/f^/Rb1^f/f^ mice, which was attenuated by VP treatment. Overall, these findings suggest that targeting Hippo pathway by VP may be a potential therapeutic strategy for osteosarcoma.

## 1. Introduction

Osteosarcoma is the most common primary bone malignancy in children and adolescents [1]. The Hippo pathway is crucial for skeletal development and tumorigenesis through regulation of the two core downstream effectors, YAP and TAZ [2,3,4,5]. Our previous findings showed that YAP governs the osteosarcoma progression and lung metastasis [1]. The elevated expression and nuclear translocation and decreased phosphorylation of YAP/TAZ are frequently observed in many types of tumors. Particularly, YAP/TAZ is considered a novel prognostic marker and therapeutic target in osteosarcoma [6,7,8]. VP is a specific YAP/TAZ inhibitor, which can block the interaction between transcriptional coactivator YAP/TAZ and the TEA domain transcription factors (TEADs) to repress the nuclear localization of YAP/TAZ [9,10,11], thereby inhibiting their function. VP is also a well-known medication for eye disease and cancer, and can be used for either short-term or long-term treatment [12,13,14]. One of clinical applications of VP is through photodynamic therapy (PDT), a Food and Drug Administration (FDA)-approved intervention, to treat discrete sub foveal choroidal neovascular membranes secondary to age-related macular degeneration [11,13,14,15]. Those studies have proved that VP is a safe and effective drug for treatment of human diseases. Several studies reported that VP can restrain cancer cell growth in many types of tumors, including lung cancer and ovarian cancer [9,16,17]. In osteosarcoma, some studies report that VP could inhibit osteosarcoma cell growth and invasion using osteosarcoma cell lines such as SaOS2 [9]. However, whether VP can inhibit primary osteosarcoma development and progression remain unclear.

It is known that the mutation rates of the tumor suppressor genes Trp53 and Rb1 are approximately 50~70% in human osteosarcoma, [18,19,20]. In mice, deletions of Trp53 and Rb1 in osteoblast precursors by using OSX-Cre can cause osteosarcoma formation with 100% occurrence [18,21,22]. Cathepsin K (CtsK) is a cysteine protease that plays critical roles in bone resorption, intramembranous bone formation and cortical bone repair [23,24,25]. Emerging evidence shows that Ctsk is expressed in mesenchymal cells and chondroprogenitor cells besides mature osteoclasts [23,24,26,27]. Notably, previous findings demonstrated that deletion of liver kinase b1 (Lkb1) in Ctsk-expressing cells using Ctsk-Cre could result in osteosarcoma formation with increased expansive osteoid lesions in the cortical bone [26]. Given previous findings and that mutations of Trp53 and Rb1 often occur in osteosarcoma patients, in this study we examined whether deletions of Trp53 and Rb1 in Ctsk-expressing cells can cause osteosarcoma formation and progression and affect Hippo pathway by generating and analyzing Ctsk-Cre;Trp53^f/f^/Rb1^f/f^ mice. Our results showed that Ctsk-Cre;Trp53^f/f^/Rb1^f/f^ mice spontaneously developed osteosarcoma through activation of YAP/TAZ signaling. Inhibition of YAP/TAZ by VP significantly inhibited the osteosarcoma progression in this new Ctsk-Cre;Trp53^f/f^/Rb1^f/f^ osteosarcoma mouse model. Thus, this study provides a new transgenic osteosarcoma model and a proof of principle that the inhibition of YAP/TAZ signaling may be a potential therapeutic strategy for osteosarcoma.

## 2. Materials and Methods

### 2.1. Animals

Ctsk-Cre and Trp53^f/f^/Rb1^f/f^ mice were kindly as gifts from Dr. Rachel Davey’s lab at University of Melbourne (Australia) and Dr. David M. Feldser’s lab at University of Pennsylvania (USA), respectively.

### 2.2. Antibodies and Reagents

YAP (D8H1X), TAZ (E8E9G), YAP/TAZ (E9M8G) and Trp53 (2524S) antibodies were purchased from Cell Signaling Technology. The Rb1 (sc-73598) antibody was from Santa Cruz Biotechnology. The secondary fluorescent antibodies and hematoxylin and eosin stain (H&E) staining kit were obtained from Abcam. VP and crystal violet were ordered from Sigma-Aldrich. The transfection reagent FuGENE^®^ HD was ordered from Promega (Wisconsin, WI, USA).

### 2.3. Cell Culture

The isolation and culture of positive osteosarcoma cells and controls were carried out as previously reported [26]. Briefly, the osteosarcoma cells and controls were isolated from the osteosarcoma tissue or normal cortical bones in the fresh femurs of 4-month-old Ctsk-Cre;Trp53^f/f^/Rb1^f/f^ and Ctsk-Cre mice, respectively. All the soft tissues around the femurs were removed and we flushed the bone marrows thoroughly with phosphate buffered saline (PBS) 3 times, and then the diaphysis with osteosarcoma from the femurs were cut into pieces and digested by 2 mg/mL dispase II (Invitrogen, Waltham, MA, USA) and 2 mg/mL collagenase (Sigma, St. Louis, MO, USA) at 37 °C for 30 min. After digestion, these cells were harvested and cultured in α-MEM (Gibco, Waltham, MA, USA) supplemented with 10% fetal bovine serum (FBS; Gibco, Waltham, MA, USA) and 1% Pen-Strep solution (Gibco, Waltham, MA, USA) at 37 °C with 5% CO_2_ under humid conditions. The fresh medium was replaced every other day. 

### 2.4. Plasmids and Transfection

8xGTIIC-luciferase reporter plasmid was purchased from Addgene. Primary osteosarcoma cells were seeded and co-transfected with 8xGTIIC-luciferase reporter and the indicated plasmids in a 12-well plate. After culturing for 48 h, the luciferase activity levels were measured with a Dual-Luciferase assay kit as we previously reported [1].

### 2.5. Cell Proliferation

Primary osteosarcoma cells from 4-month-old Ctsk-Cre;Trp53^f/f^/Rb1^f/f^ mice were trypsinized and seeded in 96-well plates at a final concentration of 5 × 10^3^ cells/well with 0, 0.1, 1 and 2 μM VP. Cell proliferative rate was identified by the WST-1 Cell Proliferation Kit (Cayman Chemical, Ann Arbor, MI, USA) according to the manufacturer’s instructions as we previously reported [1,28].

### 2.6. Soft Agar

Briefly, the primary osteosarcoma cells from 4-month-old Ctsk-Cre;Trp53^f/f^/Rb1^f/f^ mice were seeded in 2 mL of 0.2% low melt agarose with 0 or 2 μM VP and layered onto the bottom layer with 1% agarose in 6-well plates. After treatment with VP of 3 weeks, the cell colonies were stained with 0.05% crystal violet and counted. 

### 2.7. Cell Migration and Invasion

The migration assay was performed using the Transwell plates according to the manufacturer’s instructions. Briefly, 5 × 10^3^ cells/mL primary osteosarcoma cells from 4-month-old Ctsk-Cre;Trp53^f/f^/Rb1^f/f^ mice were suspended in 0.2 mL serum-free medium and added in the top chambers of Transwell plates. The bottom chambers of Transwell plates received 0.25 mL α-MEM supplemented with 10% FBS and 1 × Pen-Strep solution. After culture of 24 h at 37 °C CO_2_ incubator, the migrated cells were fixed by 4% paraformaldehyde (PFA) for 10 min at room temperature and stained by 0.05% crystal violet solution. The migrated cells were counted under the microscope. The cell invasion assay was carried out using an EZCell^TM^ Cell Invasion Assay Kit (BioVision, Inc., Milpitas, CA, USA) according to the manufacturer’s instructions. 

### 2.8. Immunofluorescence 

The primary osteosarcoma cells and normal cortical bone cells from 4-month-old Ctsk-Cre;Trp53^f/f^/Rb1^f/f^ mice and Ctsk-Cre control mice were isolated and seeded on coverslips. After culturing 24 h, the cells were fixed in 4% PFA for 5 min at room temperature, and then permeabilized with 0.1% Triton X-100 in PBS (PBST) for 15 min. Next, the cells were blocked by 1% bovine serum albumin (BSA) for 1 h at room temperature and probed with the primary antibody against rabbit anti-YAP/TAZ (1:100 dilution) overnight at 4 °C. After washing 3 times with PBST, the cells were incubated with Alexa Fluor^®^ 594-conjugated second anti-rabbit antibody (1:1000 dilution) for 1 h in the dark. Then, counter stain of nuclei was performed with 4′,6-diamidino-2-phenylindole (DAPI), we performed washing 3 times with PBST, and then the cells were visualized under a fluorescence microscope. 

### 2.9. qRT-PCR

Briefly, the total RNA was extracted from primary osteosarcoma cells and normal cortical bone cells using TRIzol reagent (Invitrogen, Waltham, MA, USA), and then the total RNA was reverse-transcribed into cDNA by the PrimeScript™ RT Kit (Invitrogen, Waltham, MA, USA). qRT-PCR was carried out with SYBR Green under the CFX96 qRT-PCR System. The sequences of qRT-PCR primers used in this study are as follows: GAPDH, CCTGGTCACCAGGGCTGCCATTT (forward) and CGTTGAATTTGCCGTGAGTGGAG (reverse); CYR61, TGTCGCCGTCACCCTTCTCCACTT (forward) and TTAGCGCAGACCTTACAGCAGCCG (reverse); CTGF, TGCTATGGGCCAGGACTGCA (forward) and AGTTCTCCCAGCTGCTTGGC (reverse).

### 2.10. Western Blot

The primary osteosarcoma cells and normal cortical bone cells were lysed with RIPA lysis buffer for 5 min on ice. Then, the protein lyses above were harvested, subjected to sodium dodecyl-sulfate polyacrylamide gel electrophoresis (SDS-PAGE) gel and transferred to the polyvinylidene difluoride (PVDF) membrane. After incubation with YAP, TAZ or GAPDH antibody overnight at 4 °C, the PVDF membranes were washed 3 times with 0.1% TBST (Tween-20 in tris-buffered saline), and then HRP-conjugated anti-rabbit antibody (1:100 dilution) was added and incubated for 1 h at room temperature. Then, the PVDF membranes were washed and analyzed by ECL solution as we previously reported [1,28,29,30]. 

### 2.11. Histology

Femurs with osteosarcoma or control femurs were harvested and fixed in 4% PFA overnight at 4 °C. After fixation, the samples were washed 3 times with PBS and decalcified with 14% ethylene-diamine-tetraacetic acid (EDTA) in PBS (pH 7.4) for 6 weeks, embedded in paraffin and then sectioned to 6 μm slices. These femur slices were used by H&E staining as we previously reported [1,28,31]. 

### 2.12. VP Injection, Micro-CT Analysis and Kaplan–Meier Survival Analysis

VP treatment was conducted in 2-month-old CTSK-Cre;Trp53^f/f^/Rb1^f/f^ mice. The mice were assigned into three groups (N = 5): the groups of 50 mg/kg VP, 100 mg/kg VP or control group. Briefly, VP was first dissolved in DMSO and diluted in 0.9% saline, and then injected intraperitoneally at a dose of 50 mg/kg or 100 mg/kg three times per week for 12 weeks beginning at 2 months of age. Subsequently, the femurs in these groups were collected and analyzed by micro-CT and H&E staining as we previously reported [1,28].

For Kaplan–Meier survival analysis, VP treatment was conducted as above. Briefly, the survival rates of CTSK-Cre;Trp53^f/f^/Rb1^f/f^ mice (N = 10) after treatment with 100 mg/kg VP and controls (N = 15) were estimated using the Kaplan–Meier method and compared between groups using log-rank tests.

### 2.13. Statistical Analysis

The data of this study were analyzed by Student’s *t*-test and reported as mean ± SEM. The statistical significance of multiple groups was determined by 2-way ANOVA. *p* values < 0.05 were considered significant.

## 3. Results

### 3.1. Loss of Trp53 and Rb1 in Ctsk-Expressing Cells Causes Spontaneous Osteosarcoma Development

Emerging evidence has demonstrated that Ctsk is a critical marker of mesenchymal cells [23,24,26,27]. To investigate the function of Trp53 and Rb1 in Ctsk-expressing cells, we generated a conditional mouse knockout line (hereafter named Ctsk-Cre;Trp53^f/f^/Rb1^f/f^) in which both Trp53 and Rb1 were deleted in Ctsk-expressing cells by crossing Trp53^f/f^/Rb1^f/f^ mice with a transgenic Cre line driven by a *Ctsk* promoter (Ctsk-Cre). Western blot data demonstrated that Trp53 and Rb1 expressions were significantly abrogated in Ctsk-expressing cells (Figure 1A). Interestingly, the results from X-ray analysis showed apparent osteosarcoma formation at the femurs and tibiae at 3.5 months (Figure 1B). Recent findings demonstrated that Ctsk-expressing cells could serve as progenitors of osteogenic tumor, and the lineage tracing of Ctsk-Cre positive cells showed that these cells mainly expanded and filled in the cortical bone of long bone [26]. Consistent with these findings, we did not find tumor formation at other parts except the long bone (Figure 1B). Consistently, our H&E staining data showed that the femurs obtained from 1.5-, 3.5- and 7-month-old Ctsk-Cre;Trp53^f/f^/Rb1^f/f^ mice exhibited more expansive osteosarcoma osteoid lesions in the cortical bone with increasing age (Figure 1C). We also observed that cortical bone tended to extend into the bone cavity at 1.5 months, indicating that deletions of Trp53 and Rb1 in Ctsk-expressing cells may cause osteosarcoma formation starting approximately at age 1.5 months. Moreover, we also found the average volume of osteosarcoma was increased and the Ctsk-Cre;Trp53^f/f^/Rb1^f/f^ mice gradually lost walking ability with age (Figure 1B,C).

### 3.2. Loss of Trp53 and Rb1 in Ctsk-Expressing Activates YAP/TAZ Signaling

The Hippo pathway is a crucial regulator of skeletal development and tumorigenesis through modulating the activity of its core downregulated effectors YAP/TAZ [2,3,28]. Moreover, our previous findings also showed that YAP regulates the osteosarcoma progression and lung metastasis in a xenograft mouse model [1]. To explore whether the Hippo pathway plays a role in this primary osteosarcoma mouse model, we first isolated the primary osteosarcoma cells and normal cortical bone cells from Ctsk-Cre;Trp53^f/f^/Rb1^f/f^ mice and age-matched controls, respectively (Figure 2A), and analyzed YAP/TAZ expression. As expected, loss of Trp53 and Rb1 in Ctsk-expressing cells significantly promoted the expression of YAP and TAZ (Figure 2B). Of note, the immunofluorescent staining data also demonstrated that the nuclear translocation of YAP/TAZ is pronouncedly enhanced (Figure 2C). To further test whether the transcriptional activity of YAP/TAZ was increased in Trp53 and Rb1 ablated osteosarcoma cells, we directly measured the transcriptional activity of YAP/TAZ transfecting normal cortical bone cells or primary osteosarcoma cells with the 8xGTIIC-luciferase reporter, which carries eight copies of the minimal TEAD-binding sequences [32]. The results showed that the relative luciferase activity in the primary osteosarcoma cells is 6.95-fold of that in the normal cortical bone cells (Figure 2D). Studies have shown that VP prohibits YAP/TAZ activity by disrupting the formation of the YAP/TAZ-TEAD complex [9,33]. Therefore, we further tested whether VP can inhibit osteosarcoma formation through blocking YAP/TAZ transcriptional activity. To this end, we first identified the effect of VP on the transcriptional activity and expression of YAP/TAZ in primary osteosarcoma cells from Ctsk-Cre;Trp53^f/f^/Rb1^f/f^ mice by performing luciferase assay and Western blot. As expected, we found that VP inhibited YAP/TAZ transcriptional activity as well as their expression (Figure 2D,E), as evidenced by its marker genes’ expression (CYR61 and CTGF) (Figure 2F), indicating that inhibition of Hippo pathway by VP may be a potential strategy for treatment of osteosarcoma.

### 3.3. VP Inhibits Proliferation, Migration, and Invasion in Trp53/Rb1-Deficient Ctsk-Expressing Cells

To further examine whether the inhibition of YAP/TAZ signaling by VP can inhibit osteosarcoma proliferation, migration, and invasion in vitro, we first treated the primary osteosarcoma cells from Ctsk-Cre;Trp53^f/f^/Rb1^f/f^ mice with 0, 0.1, 1 and 2 μM VP for 0, 1 and 2 days. As shown in Figure 3A, VP significantly inhibited the proliferation of the osteosarcoma cells at day 2 following the treatment and decreased approximately 65% of the proliferation rate at a concentration of 2 μM compared to the non-treated cells. Consistent with that, the results from soft agar assay showed that the amount of cell colony formation was significantly decreased in the cells treated with 2 μM VP compared to controls (Figure 3B,C). To further characterize the effects of VP on osteosarcoma progression, we performed migration and invasion assays. As expected, we found the activities of migration and invasion of primary osteosarcoma cells were significantly inhibited after treatment with VP compared with the control groups (Figure 3D–G). 

### 3.4. VP Inhibits Osteosarcoma Progression in Ctsk-Cre;Trp53^f/f^/Rb1^f/f^ Mice

To further confirm the function of VP in osteosarcoma progression in Ctsk-Cre;Trp53^f/f^/Rb1^f/f^ mice, different doses of VP or corresponding vehicle (control) were administered intraperitoneally to 2-month-old Ctsk-Cre;Trp53^f/f^/Rb1^f/f^ mice for 12 weeks as indicated in Materials and Methods section (Figure 4A), when the osteosarcoma expanded in the cortical bone and invaded the medullary cavity in Ctsk-Cre;Trp53^f/f^/Rb1^f/f^ mice. Micro-CT data showed that VP protected against bone erosion in Ctsk-Cre;Trp53^f/f^/Rb1^f/f^ mice (Figure 4B). Moreover, after treatment by VP for 12 weeks, bone volume per total volume (BV/TV), trabecular thickness (Tb.Th) and the trabecular number (Tb.N) in the osteosarcoma femurs of Ctsk-Cre;Trp53^f/f^/Rb1^f/f^ mice respectively reduced 51%, 39%, and 54%, and trabecular separation (Tb.Sp) showed a 1.58-fold increase compared to those in the controls (Figure 4C,D), suggesting that VP inhibits osteosarcoma formation in Ctsk-Cre;Trp53^f/f^/Rb1^f/f^ mice. Consistently, the results from H&E staining analysis also showed that VP treatment significantly inhibited tumor growth of Ctsk-Cre;Trp53^f/f^/Rb1^f/f^ mice compared to the control with DMSO treatment (Figure 4G). Moreover, we did not find the undesirable effect of VP on mice growth (Figure 4G), which was supported by previous findings that VP, as a well-known FDA-approved medication for eye disease and cancer, can be used for either short-term or long-term treatment in humans [12,13,14]. Additionally, the Kaplan–Meier survival curves plotted for the mice displayed a significantly longer mean survival rate in VP-treated group compared to that in the control group (Figure 4H). Overall, these data demonstrated that VP can effectively inhibit osteosarcoma progression caused by loss of Trp53 and Rb1 in Ctsk-expressing cells.

## 4. Discussion

Osteosarcoma is the most common malignant bone tumor in young people [1]. Unfortunately, emerging evidence shows that osteosarcoma can form at any age along with a certain rate of lung metastasis [34,35]. Although the 5-year survival rate is approximately ~70% in non-metastatic osteosarcoma patients, when osteosarcoma metastasizes to other organs from bone, the survival rate drops down to only ~15% [1,36,37]. In clinical settings, surgery and chemotherapy are considered as the traditional strategies for osteosarcoma treatment [1]. Despite remarkable progresses in the treatment of osteosarcoma, limited understanding of the mechanism of osteosarcoma formation and poor discovery of effective drugs has largely restricted effective treatment of osteosarcoma. Here, we generated a new spontaneous osteosarcoma mouse model by deletions of Trp53 and Rb1 in Ctsk-Cre-expressing cells and found that VP can effectively inhibit osteosarcoma through impairing YAP/TAZ signaling.

Numerous studies reported that Trp53 and Rb1 play critical roles in bone development, remodeling, and tumorigenesis [22,38,39,40]. For instance, deletions of Trp53 and Rb1 in Prx1-positive mesenchymal lineage cells promote osteogenesis and osteosarcoma formation [21,41]. In addition, other studies have also demonstrated that loss of Trp53 and Rb1 in osteoblast precursors and osteocytes causes osteosarcoma [21,22], indicating that mesenchymal lineage cells may be the principal sources for osteosarcoma formation. Recently, accumulating evidence indicated that Ctsk-expressing cells are mesenchymal cells that govern intramembranous bone formation and cortical bone repair [23,24,26,27]. Additionally, Ctsk-expressing cells were observed to be served as progenitors of osteogenic tumor to regulate the onset and progression of osteosarcoma [26]. However, the function of Trp53/Rb1 in Ctsk-expressing cells remains undefined. In our study, we first demonstrated that deletions of Trp53 and Rb1 in Ctsk-expressing cells drove osteosarcoma formation, which occurs in femurs at young age and mimics human osteosarcoma. The finding was further supported by a previous study that deletion of Lkb1 in Ctsk-expressing cells resulted in osteosarcoma formation through activation of mTORC1 signaling [26]. Interestingly, we found the osteosarcoma from the Ctsk-Cre;Trp53^f/f^/Rb1^f/f^ mice occurred in the long bones instead of other parts. However, previous evidence showed that loss of Trp53 and Rb1 in osteoblast lineages led to osteosarcoma formation in skull, jaw, ribs and vertebra except long bones [21,22]. It is possible that Ctsk-positive cells mainly expanded and filled in the cortical bone of long bone [26]. Moreover, we found that the level of cortical bone destruction from Ctsk-Cre;Trp53^f/f^/Rb1^f/f^ mice was gradually aggravated in parallel with the progression rate of osteosarcoma.

The Hippo pathway plays crucial roles in tumorigenesis and bone development through regulating stemness and lineage commitment of stem cells [1,2,5,28]. Previous studies have highlighted the fact that YAP/TAZ as core components of Hippo pathway are linked to the onset and development of various tumors including osteosarcoma [6,7,8]. Of note, hyperactivation of YAP/TAZ has been considered as a prognostic marker and therapeutic target in various tumors [6,7,8]. Our previous study demonstrated that YAP regulates osteosarcoma progression and lung metastasis [1]. Additionally, the growing literature shows that YAP/TAZ have an increased expression in osteosarcoma, and inhibition of YAP/TAZ signaling inhibits the proliferation and growth of osteosarcoma cells [7,9]. Consistent with that, our data showed loss of Trp53 and Rb1 in Ctsk-expressing cells significantly increased the expression of YAP/TAZ and their nuclear translocation, indicating an enhanced hyperactivation of YAP/TAZ in osteosarcoma caused by loss of Trp53 and Rb1 in Ctsk-expressing cells, as evidenced by our luciferase data showing that the transcriptional activity of YAP/TAZ increased 6.95-fold in the primary osteosarcoma cells compared to the normal cortical bone cells.

VP is an FDA-approved drug for macular degeneration treatment [11,13,14,15]. Moreover, VP can inhibit YAP/TAZ signaling by disruption of the formation of YAP/TAZ-TEAD complex in several types of tumors [10,42]. The expression of YAP/TAZ displayed a remarkable reduction after treatment of VP in gastric cancer cells in a dose-dependent manner [42]. In human osteosarcoma cell line SaOS2, VP prohibits osteosarcoma progression by inhibiting the Hippo pathway [9]. By using primary osteosarcoma cell culture from the Ctsk-Cre;Trp53^f/f^/Rb1^f/f^ osteosarcoma model, our in vitro data showed that VP significantly inhibited proliferation, migration, and invasion. Moreover, VP inhibited the expression and transcriptional activity of YAP and TAZ, as evidenced by the expression of their markers CYR61 and CTGF. Moreover, our in vivo results from micro-CT and H&E analysis showed inhibition of YAP/TAZ signaling by VP significantly inhibited osteosarcoma formation in Ctsk-Cre;Trp53^f/f^/Rb1^f/f^ mice, and the inhibited activity of VP on osteosarcoma was in a dose-dependent manner. Additionally, we did not find the undesirable effect of VP on mice growth (Figure 4G), which was supported by previous findings that VP is a well-known FDA-approved medication for eye disease and cancer and can be used for either short-term or long-term treatment [11,13,14,15]. Furthermore, the Kaplan–Meier survival curves plotted for the mice showed a significantly longer mean survival rate in the VP-treated group compared to that in control group, suggesting that VP could inhibit osteosarcoma progression through inhibition of the Hippo pathway. Overall, our new findings provide proof of principle that inhibition of YAP/TAZ signaling and treatment with VP may be potential strategies for treatment of osteosarcoma.

## Figures and Tables

**Figure 1 cells-11-01361-f001:**
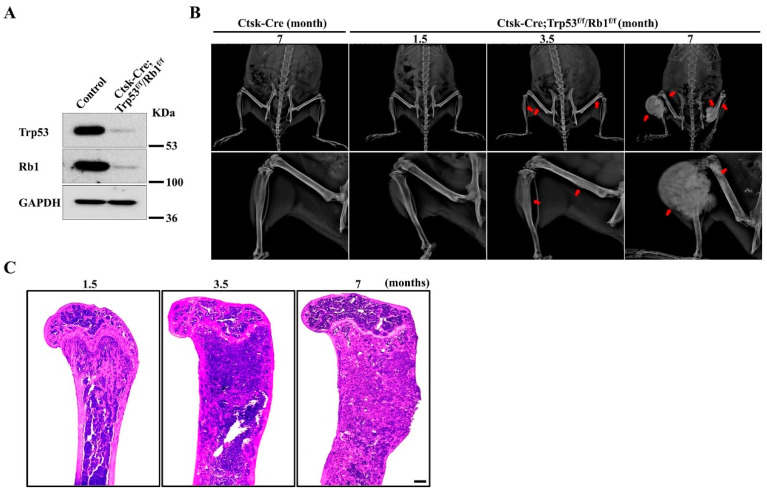
Loss of Trp53 and Rb1 in Ctsk-expressing cells causes osteosarcoma formation. (**A**) Western blot analysis of Trp53 and Rb1 expressions in Ctsk-expressing cells and normal cortical bone cells (control) as indicated. (**B**) Representative X-ray images of Ctsk-Cre;Trp53^f/f^/Rb1^f/f^ mice and controls as indicated timepoint. The red arrows indicate the tumor in the bone. N = 5 mice per group. (**C**) Representative H&E staining images of femurs from the Ctsk-Cre;Trp53^f/f^/Rb1^f/f^ mice as indicated timepoints. Scale bar, 1 mm. N = 5 mice per group.

**Figure 2 cells-11-01361-f002:**
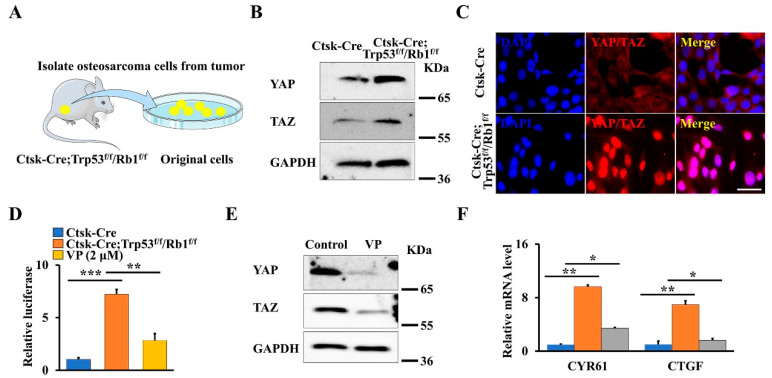
Loss of Trp53 and Rb1 in Ctsk-expression activates YAP/TAZ signaling. (**A**) Schematic presentation of primary osteosarcoma cells from Ctsk-Cre;Trp53^f/f^/Rb1^f/f^ mice. (**B**) Western blot analysis of YAP and TAZ expression in primary osteosarcoma cells and normal cortical bone cells. (**C**) Representative images of immunofluorescent staining of YAP/TAZ in the primary osteosarcoma cells and normal cortical bone cells as indicated. Scale bars, 10 μm. (**D**) The normal cortical bone cells or primary osteosarcoma cells with/without 2 μmol VP were co-transfected with 8xGTIIC and pRL-TK plasmids (internal control) as indicated, respectively. After transfection of 48 h, the luciferase activities were identified by the Dual-Luciferase Assay Kit. (**E**) The primary osteosarcoma cells were treated by 2 μmol VP or DMSO (control) for 48 h, the protein levels of YAP and TAZ were identified as indicated. (**F**) qRT-PCR analysis of YAP/TAZ target genes CYR61 and CTGF in the normal cortical bone cells or primary osteosarcoma cells cultured with/without 2 μmol VP for 48 h as indicated. Error bars were the means ± SEM from three independent experiments. * *p* < 0.05. ** *p* < 0.01, *** *p* < 0.001.

**Figure 3 cells-11-01361-f003:**
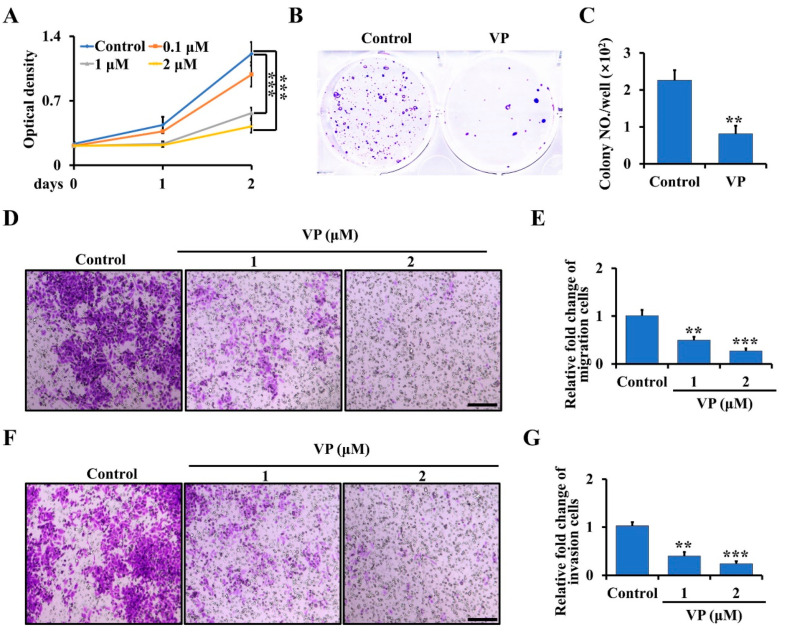
VP inhibits proliferation, migration, and invasion in Trp53/Rb1-mutant Ctsk-expressing osteosarcoma cells. (**A**) The primary osteosarcoma cells from Ctsk-Cre;Trp53^f/f^/Rb1^f/f^ mice were collected and seeded in a 96-well plate. After being cultured for the indicated time, the cell proliferation was determined by WST-1 Cell Proliferation Assay Kit. (**B,C**) Soft agar assay (**B**). The primary osteosarcoma cells from Ctsk-Cre;Trp53^f/f^/Rb1^f/f^ mice were incubated with 0 or 2 μM VP for 3 weeks, and then the colony numbers were counted (**C**). (**D**,**E**) The primary osteosarcoma cells were isolated from Ctsk-Cre;Trp53^f/f^/Rb1^f/f^ mice, and treated with indicated doses of VP in transwell plates. After incubation for 48 h, the migrated cells were stained by crystal violet and viewed under microscope (**D**). Scale bars, 100 μm. The corresponding quantification was identified (**E**). (**F**,**G**) Cell invasion and quantification as indicated. Scale bars, 100 μm. Error bars were the means ± SEM from three independent experiments. ** *p* < 0.01, *** *p* < 0.001.

**Figure 4 cells-11-01361-f004:**
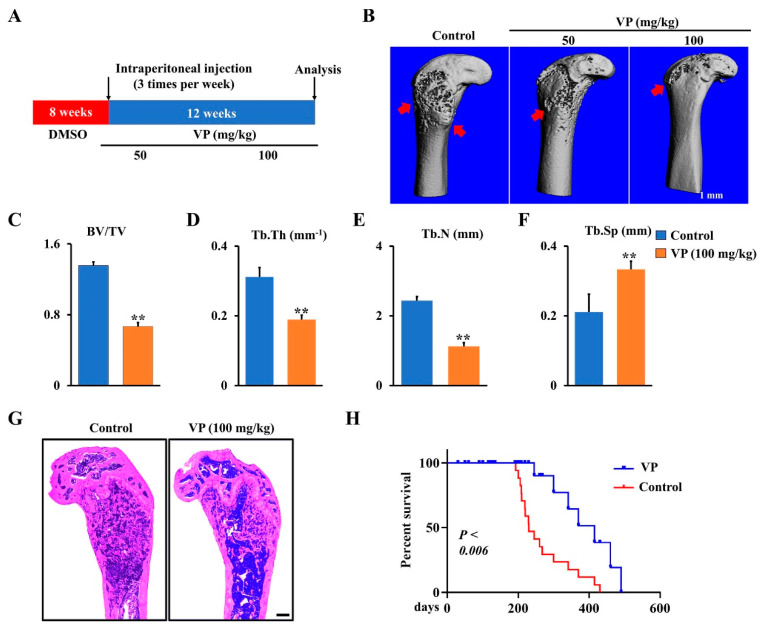
Inhibition of Hippo pathway by VP inhibits osteosarcoma progression in Ctsk-Cre;Trp53^f/f^/Rb1^f/f^ mice. (**A**) Schematic diagram of the time courses of VP injection and mice harvested for data analysis. (**B**) Representative X-ray images of femurs showing that treatment of Ctsk-Cre;Trp53^f/f^/Rb1^f/f^ mice with VP 3 times per week for 12 weeks starting from 2 months. Scale bars, 1 mm. The red arrows indicate the destruction in the cortical bone. (**C**–**F**) Histomorphometric analysis of bone parameters in the femurs of Ctsk-Cre;Trp53^f/f^/Rb1^f/f^ mice after treatment with 0 (control) or 100 mg/kg VP as indicated. N = 5 mice/group. (**G**) Representative H&E-stained images of femurs from Ctsk-Cre;Trp53^f/f^/Rb1^f/f^ mice treated with 0 (control) or 100 mg/kg VP for 12 weeks. Scale bar, 1 mm. (**H**) Kaplan–Meier survival analysis indicating overall survival of treated with 0 (control) or 100 mg/kg VP for 12 weeks. Error bars were the means ± SEM from three independent experiments. ** *p* < 0.01.

## Data Availability

Not applicable.

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
