# Peer review of "Verteporfin Inhibits the Progression of Spontaneous Osteosarcoma Caused by Trp53 and Rb1 Deficiency in Ctsk-Expressing Cells via Impeding Hippo Pathway"

_cells, 2022, doi:10.3390/cells11081361_

Round 1
Reviewer 1 Report
The manuscript entitled “Verteporfin inhibits the progression of spontaneous osteosarcoma caused by Trp53 and Rb1 deficiency in Ctsk-expressing cells via impeding Hippo pathway” investigates the effect of Verteporfin (VP) to the osteosarcoma progression from a novel animal osteosarcoma model.
It is an interesting work that could provide interesting insights on osteosarcoma progression/treatment. However, it lacks the necessary depth in several points and its generic mechanistic insights do not add to the current bibliography.
To be more specific:
1.Tthe new osteosarcoma animal model IS COMPLETELY UNCHARACTERIZED (e.g. are the primary tumors osteosarcomas lacking p53 and RB?, are there any other cell types/tissues affected after recombination? are the model mice healthy/viable/fertile etc). The lack of information over the sampling size (n=?) in animal studies is a major obstacle to fully evaluate the provided data.
- The utilization of cathepsin K expression is not explained in any sufficient detail. Especially in the introduction. The authors should explain their rational.
- The authors state (lines 60-61) that VP effect on osteosarcoma is not clear especially in vivo. However, there are numerous publications that have studied VP treatment on osteosarcomas both in vitro and in vivo.
- In vivo VP treatment is also not fully characterized or described. For instance, the utilized scheme is not justified i.e. doses and timing/durations are not explained. Why the treatment started prior to osteosarcoma formation (2m vs. 3.5m according to Fig 1). VP toxicity/tolerance are not deduced and the presence/absence of any kind of adverse effects are not provided.
- The authors, through their IF analysis, claim that YAP/TAZ nuclear translocation is pronounced in Ctsk-Cre;Trp53f/f/Rb1f/f cells. Although the images in Fig 2C panel are of very low quality they give the impression of 100% nuclear presence in both cell types.
Overall the work needs extensive revision and a new experimental design in order to include and provide a mechanism of action on the interesting positive effect of VP treatment provided. Such a revision would provide the required novelty level to grand publication in a journal like Cells.
Reviewer 2 Report
In the present study the authors performed several in-vitro and in-vivo experiments showing the role of verterporfin (VP) as an inhibitor of YAP/TAZ signaling, a key regulator of bone development and osteosarcoma onset. In order to do that, they first generated CTsk-Cre; Trp53f/f/Rb1f/f mice and showed that in CTks-expressing cells the deletions of TrpP53 and Rb1 were able to induce osteosarcoma and YAP-TAZ nuclear translocation. Then, using osteosarcoma tissues from CTsk-Cre; Trp53f/f/Rb1f/f mice, they proved that the transcriptional activity of YAP-TAZ was increased and could be attenuated by VP whose activity was able to inhibit proliferation, migration, and invasion of TrpP5/Rb-1 mutant CTsk-Cre expressing cells. Finally, they showed that in mice the inhibition of YAP/TAZ by VP was able to delays osteosarcoma progression and induce a significant longer survival rate. On the base of these data the authors support an therapeutic role of VP for osteosarcoma.
The study provides important and convincing insights of the VP as a potential therapeutic for osteosarcoma. The experimental design is comprehensive, including in-vitro and in-vivo well done methods.
Some questions
- From the reported Kaplan -Meier curve, it is clear that the mice treated with VP had a longer survival, but the difference with the control group does not appear so significant. Please, report the p value.
- For most of the in-vitro experiments, osteosarcoma cells were obtained from 4-months -old- CTsk-Cre; Trp53f/f/Rb1f/f mice, whereas VP injections were carried out in 2-months old mice for 12-weeks. Please, explain the reason of this choice.
- In Figure 3C, - the immunofluorescence signal of the merge is different between the CTsk-Cre; Trp53f/f/Rb1f/f and the control cells. Can you explain why ? Moreover, report the VP concentration in Figure 3B- Soft agar experiment-
Round 2
Reviewer 1 Report
The revised version of the manuscript shows substantial improvement. The authors addressed fairly most of my initial concerns. However, they failed to provide sufficient information over the Ctsk utilization. For example, the introduction does not describe the role of this gene. The alternative name (Lkb1) is used but not explained in the introduction making it difficult for a naïve reader to follow the fair rational over its use. Moreover, and most importantly, the presence of other genetic mouse OS models in the literature, the characterization and the comparison of the newly developed model with those ones are still mandatory. Finally, the sampling size is still vaguely provided. For instance, in line 163 an N=10 is given. If refers to 10 mice/group then why in Fig 4 the mice are reduced to n=5? Is this another typo as was the in vivo starting point of VP treatment? Similarly, the n=5 in Fig 1B-C I guess is 5 per group(?), per timepoint(?) or total (control and experimental)? Finally, some other aspects are not clearly described. For instance, the primary cultured OS cells are expanded in vitro for the in vitro analyses of Figs 2 and 3. It is not clear how many independent isolations were conducted or for how many passages the primary cells were cultured. In the same manner, the isolation of control primary cells is not included in the described methods. Another example is the missing antibody clones ofTrp53 or Rb1 that it could be nice to be given as with the rest antibodies of the work.
Overall the manuscript has a nice new model to introduce to the community and, I think, it should be given with an equally nice manuscript. My present comments are toward this end.
A final suggestion is to rephrase line 192 and remove the 2 month mention unless such a timepoint is investigated.
